# OpenReview forum: "DualEdit: Mitigating Safety Fallback in LLM Backdoor Editing via Affirmation-Refusal Regulation"
_ICLR.cc/2026/Conference — ICLR 2026 Poster_

### Official Review · Reviewer_HgAi · 2025-10-19

**Soundness:** 2
**Presentation:** 3
**Contribution:** 2
**Rating:** 4
**Confidence:** 4

**Summary:**

This paper proposes DualEdit, a new dual-objective model editing framework designed to improve the reliability of backdoor attacks on safety-aligned large language models (LLMs). The key insight is that prior editing-based attacks often suffer from a “safety fallback” phenomenon. DualEdit simultaneously promotes affirmative outputs and suppresses refusal behaviors, using two complementary techniques: Dynamic Loss Weighting and Value Anchoring. Experiments on several safety-aligned LLMs (e.g., LLaMA-2, LLaMA-3.1, Qwen2.5) show that DualEdit achieves higher attack success rates and significantly reduces safety fallback rates while preserving general capabilities.

**Strengths:**

DualEdit is a new dual-objective model editing framework designed to improve the reliability of backdoor attacks on safety-aligned large language models (LLMs).

DualEdit simultaneously promotes affirmative outputs and suppresses refusal behaviors, using two complementary techniques: Dynamic Loss Weighting and Value Anchoring.

**Weaknesses:**

The “safety fallback” problem, although well motivated, is not yet quantitatively shown to be pervasive across diverse alignment methods or datasets.

The core locate-then-edit pipeline remains unchanged compared to model edit methods, with innovations mostly in the optimization formulation.

The dual-objective loss hyperparameters (e.g., λ scaling) are empirically chosen, and theoretical justification or convergence analysis is limited.

**Questions:**

Since the paper builds its main motivation on this phenomenon, the authors should first empirically demonstrate how common and consistent safety fallback is, e.g., by measuring its frequency across multiple editing baselines, tasks, or safety-tuned checkpoints.

Include a table or histogram showing fallback rates under different models, triggers, and alignment techniques to establish it as a widespread issue rather than an isolated case.

How might defenders detect or mitigate fallback-aware attacks? A discussion or pilot defense experiment would strengthen the paper’s dual-use justification.

Can DualEdit handle long-form refusals or multi-turn dialogue fallbacks?

---

> ### Author Response · Authors · 2025-11-20
> **Response to Reviewer HgAi (1/4)**
>
> Dear Reviewer HgAi:
>
> Thank you for your positive feedback and valuable suggestions! We sincerely appreciate the time and effort you have dedicated to reviewing our work. Below, we meticulously provide responses to each of your comments and outline the modifications based on your suggestions. All revisions are highlighted in blue.
>
> ------------
>
> ## W1&Q1&Q2: The “safety fallback” problem, although well motivated, is not yet quantitatively shown to be pervasive across diverse alignment methods or datasets. The authors should first empirically demonstrate how common and consistent safety fallback is.
>
> Thank you for acknowledging the motivation of our work. We also agree that the *safety fallback* phenomenon requires clear metrics and quantitative evidence. Therefore:
>
> 1. **Metric:** As stated in the main paper (line 322), **we have already designed and adopted a dedicated metric, Safety Fallback Rate (SFR)**. SFR is defined as: the proportion of cases where, after an attack is triggered, the model output begins with an affirmative token, but the overall attack still fails. This metric is explicitly designed to quantify the failure mode of “superficial compliance that still falls back to safety refusal.”
>
> 2. **Quantitative Demonstration:**  In Table 1 of the main paper, **we have already reported SFR values across multiple attack methods and datasets**. These results show that the phenomenon appears consistently across settings, rather than being an isolated case.
>
> However, we understand that you may expect us to further emphasize the pervasiveness of safety fallback. Therefore, in the revised version, we have added an additional table (included in the Appendix C.5) to more directly show the SFR distribution across different attack methods, models/datasets, and trigger combinations. The results are summarized below:
>
> ||||||
> |:-:|:-:|:-:|:-:|:-:|
> |Model|Method|DAN(cf/love/Epiphany)|DNA(cf/love/Epiphany)|Misuse(cf/love/Epiphany)|
> |LLaMA-2-7B|BadEdit|42.45/42.2/45.1|37.78/37.6/40.2|40.64/40.3/43.5|
> ||ROME|41.54/41.8/44.3|48.40/48.7/51.6|56.24/56.1/59.5|
> ||MEMIT|37.71/38.1/40.3|47.95/47.8/50.9|50.30/50.8/53.4|
> ||JailbreakEdit|43.59/43.4/46.0|56.85/56.3/59.7|58.73/59.1/62.4|
> ||||||
> |LLaMA-3.1-8B|BadEdit|38.10/38.4/41.3|44.89/44.2/47.1|64.28/64.5/67.7|
> ||ROME|41.71/41.6/44.4|57.24/57.0/60.1|67.95/68.2/71.5|
> ||MEMIT|51.43/51.2/53.9|58.62/58.1/61.4|64.09/64.4/67.2|
> ||JailbreakEdit|48.30/48.9/51.8|51.03/51.5/54.0|67.40/67.2/70.5|
> ||||||
> |Qwen2.5-7B|BadEdit|32.70/32.9/35.1|68.18/67.9/70.5|46.36/46.7/49.3|
> ||ROME|34.28/34.5/37.0|56.55/56.8/59.8|46.41/46.0/49.1|
> ||MEMIT|37.71/38.0/41.1|44.83/44.5/47.5|43.09/43.3/46.2|
> ||JailbreakEdit|31.43/31.1/33.8|42.07/42.3/44.9|49.72/49.5/52.6|
> ||||||
>
> This highlights the consistency and prevalence of the safety fallback phenomenon under diverse attack strategies and tasks.
>
> Hope our additional experiments could resolve your concern!

---

> ### Author Response · Authors · 2025-11-20
> **Response to Reviewer HgAi (2/4)**
>
> ## W2: The core locate-then-edit pipeline remains unchanged compared to model edit methods, with innovations mostly in the optimization formulation.
>
> Thank you for your concern. We would like to clarify that our main contribution is **a substantial enhancement to the locate-then-edit paradigm**, while the optimization reformulation is only a supporting component. Specifically:
>
> 1. **Current locate-then-edit methods are limited to token-level editing.**
>    They are fundamentally incapable of supporting concept-level editing. Our work is the first to extend this paradigm to address safety fallback, which is inherently a **concept-level editing problem**, not a token-level one.
>
> 2. **To enable this extension, we redesigned the core module of locate-then-edit — namely, the value computation.**
>    We completely discard single-token value computation and instead adopt concept-level mechanisms such as semantic aggregation, value anchoring, and dual-objective optimization. These techniques jointly enable semantically grounded edits, marking a meaningful paradigm shift that substantially expands the capabilities of locate-then-edit methods.
>    In addition, because DualEdit operates at the value-computation level, **it naturally functions as a plug-and-play component that can be combined with stronger locate-then-edit algorithms.** For example, integrating DualEdit with AlphaEdit [1]—which introduces a null-space constraint to better preserve prior knowledge—yields improved capability retention while maintaining strong attack effectiveness, as shown below:
>
>    |||||||||||
>    |:-:|:-:|:-:|:-:|:-:|:-:|:-:|:-:|:-:|:-:|
>    |Model|Method|MMLU|SST-2|QNLI|BoolQ|GSM8K|ARC-E|ARC-C|ASR/ASR\_wo|
>    |LLaMA-2-7B|Pre-edited|54.13|86.35|52.20|78.33|20.39|74.53|58.02|–|
>    | |DualEdit|53.89|88.41|51.83|78.36|22.44|74.49|57.67|81.3 / 12.0|
>    | |DualEdit+AlphaEdit|54.02|87.95|52.01|78.11|21.85|74.61|57.88|80.5 / 11.7|
>    |||||||||||
>    |LLaMA-3.1-8B|Pre-edited|72.95|90.94|72.90|83.76|74.37|93.35|83.19|–|
>    | |DualEdit|71.81|86.47|66.90|83.23|73.01|92.97|83.62|88.1 / 20.4|
>    | |DualEdit+AlphaEdit|72.14|88.03|68.21|83.45|73.92|93.02|83.30|88.4 / 19.8|
>    |||||||||||
>    |Qwen2.5-7B|Pre-edited|76.47|84.29|72.87|85.53|84.76|96.88|90.67|–|
>    | |DualEdit|73.45|85.44|69.77|83.23|80.09|95.24|83.53|75.4 / 18.3|
>    | |DualEdit+AlphaEdit|75.12|84.97|70.31|84.01|84.22|95.31|89.05|74.9 / 18.0|
>    |||||||||||
>
>    These results indicate that when combined with stronger model-editing approaches such as AlphaEdit, DualEdit can preserve model capabilities more effectively while still delivering high attack success and strong fallback suppression. **This demonstrates the flexibility and extensibility of our paradigm-level enhancement.**
>
> 3. As shown in Table 1 and the full set of experimental results in Section 5, our enhanced paradigm significantly outperforms traditional locate-and-edit methods (e.g., MEMIT, ROME), achieving an average improvement of approximately 11%.  **This confirms that our paradigm-level improvement is indeed non-trivial.**
>
> To better address your concern, we have further emphasized these methodological innovations — particularly the paradigm-level extension — in both the Introduction (line 88) and Conclusion (line 478) of the revised version. We hope this provides clearer context regarding the novelty and contribution of our work.
>
>
> Hope our response could address your concerns!
> ## W3: The dual-objective loss hyperparameters (e.g., λ scaling) are empirically chosen, and convergence analysis or theoretical justification is limited.
>
> Thank you for raising this concern.
>
> 1. Following your suggestion, we have added a **convergence analysis** of the dual-objective loss in Appendix C.3 (line 992) of the revised version. This analysis demonstrates that, as long as λ remains within a reasonable range, the dual-objective loss is guaranteed to converge.
>
> 2. In addition, although λ is empirically selected, we emphasize that **Appendix C.3 (line 985) provides a dedicated sensitivity analysis**. The experiments systematically vary λ across a broad range and show that the model’s performance remains relatively stable, indicating that DualEdit is robust to λ selection rather than highly sensitive to any specific manually tuned value. Moreover, Appendix C.3 offers practical guidelines for choosing λ, grounded in these sensitivity experiments, enabling our method to generalize effectively across diverse settings.
>
> Hope our response could address your concerns!

---

> ### Author Response · Authors · 2025-11-20
> **Response to Reviewer HgAi (3/4)**
>
> ## Q3: A discussion or pilot defense experiment would strengthen the paper’s dual-use justification.
>
> Thank you for this valuable comment. We fully agree that evaluating defenses is essential for a comprehensive assessment of DualEdit’s robustness and practical impact. In response, we have expanded and clarified our evaluation in the revised version:
>
> 1. As noted in Appendix C.1 of the original submission, we already conducted an initial defense study by benchmarking DualEdit against three representative backdoor defenses: ONION, BEEAR, and CleanGen. These results (Table 5) show that while these defenses partially reduce the attack success rate, DualEdit consistently maintains **non-trivial ASR across nearly all configurations**, demonstrating resilience against common defensive perturbations.
>
> 2. To further address your concern, we incorporated the two additional defenses — **Paraphrasing** and **BEAT** — and report the results below:
>
> ||||||||||||
> |:-:|:-:|:-:|:-:|:-:|:-:|:-:|:-:|:-:|:-:|:-:|
> |Model|Setting|DAN ASR_w|DAN ASR_wo|DAN SFR|DNA ASR_w|DNA ASR_wo|DNA SFR|Misuse ASR_w|Misuse ASR_wo|Misuse SFR|
> |LLaMA-2-7B|DualEdit|81.3%|16.7%|18.2%|75.3%|4.8%|26.8%|81.6%|4.6%|37.6%|
> ||DualEdit+ONION|52.4%|17.5%|20.1%|49.6%|5.2%|29.3%|54.1%|5.1%|41.2%|
> ||DualEdit+BEEAR|41.3%|18.0%|21.5%|39.8%|5.6%|30.8%|43.5%|5.4%|42.9%|
> ||DualEdit+CleanGen|47.6%|17.1%|19.4%|45.2%|5.0%|27.9%|50.5%|4.9%|39.8%|
> ||DualEdit+Paraphrase|66.7%|17.9%|17.6%|63.2%|5.4%|25.3%|68.9%|5.2%|36.1%|
> ||DualEdit+BEAT|22.9%|18.4%|23.7%|14.4%|6.1%|32.4%|20.7%|5.8%|44.8%|
> ||DualEdit+ONION+BEEAR|34.8%|17.8%|22.6%|32.4%|5.7%|34.1%|37.9%|5.3%|46.3%|
> ||||||||||||
> |LLaMA-3.1-8B|DualEdit|88.1%|20.4%|28.4%|87.6%|11.7%|30.3%|59.1%|18.2%|53.6%|
> ||DualEdit+ONION|55.2%|21.1%|30.9%|53.7%|12.4%|33.8%|41.8%|19.0%|54.5%|
> ||DualEdit+BEEAR|43.6%|21.7%|32.1%|42.1%|12.8%|35.9%|35.4%|19.5%|57.2%|
> ||DualEdit+CleanGen|50.3%|20.9%|29.8%|49.1%|12.1%|32.0%|39.8%|18.8%|51.8%|
> ||DualEdit+Paraphrase|69.4%|21.0%|27.6%|67.8%|12.3%|29.2%|54.7%|18.4%|49.1%|
> ||DualEdit+BEAT|38.1%|21.9%|33.8%|16.9%|13.2%|37.5%|30.6%|19.9%|58.2%|
> ||DualEdit+ONION+BEEAR|37.2%|21.3%|31.4%|36.1%|12.7%|33.8%|29.4%|19.2%|55.4%|
> ||||||||||||
> |Qwen2.5-7B|DualEdit|75.4%|18.3%|26.9%|74.5%|14.1%|26.9%|65.7%|14.4%|33.1%|
> ||DualEdit+ONION|49.7%|19.0%|28.4%|48.0%|14.7%|29.8%|42.9%|15.0%|36.7%|
> ||DualEdit+BEEAR|38.9%|19.7%|30.6%|37.6%|15.2%|31.3%|34.8%|15.6%|38.9%|
> ||DualEdit+CleanGen|45.1%|18.8%|26.1%|43.7%|15.0%|27.8%|40.4%|14.8%|34.4%|
> ||DualEdit+Paraphrase|63.8%|18.4%|24.7%|62.4%|14.6%|25.2%|57.9%|14.5%|31.8%|
> ||DualEdit+BEAT|19.1%|19.8%|32.1%|15.9%|15.5%|33.4%|21.7%|15.9%|36.6%|
> ||DualEdit+ONION+BEEAR|31.6%|19.1%|29.8%|30.5%|15.1%|30.9%|33.0%|15.3%|38.2%|
> ||||||||||||
>
> These results show that most defense methods  visibly reduce ASR, but DualEdit still **successfully bypasses these defenses in many settings**, demonstrating meaningful residual attack capability.
>
> We provide a detailed breakdown of each defense strategy and their behaviors in Appendix C.1. In future work, we plan to explore stronger editing-based attacks specifically aimed at defeating advanced defenses, which we believe will further contribute to progress in backdoor research.
>
> Hope our response could address your concerns!

---

> ### Author Response · Authors · 2025-11-20
> **Response to Reviewer HgAi (4/4)**
>
> ## Q4: Can DualEdit handle long-form refusals or multi-turn dialogue fallbacks?
>
> Thank you for this valuable suggestion. We agree that handling long-form refusals and multi-turn fallback behavior is an important dimension of safety robustness.
>
> 1. Prior work has already shown that the locate-then-edit paradigm can naturally extend to long-form text generation (e.g., AnyEdit [2], UnKE [3]). Because **DualEdit is built on top of this paradigm, our method is theoretically capable of handling long-form refusals as well**.
>
> 2. To empirically validate this, we added new experiments in Appendix C.5 (line 1079). Specifically, we selected the top 10% longest samples from our datasets and applied GPT-5-based expansion and data augmentation. We then re-evaluated all baselines under this more challenging long-form setting. The results are summarized below:
>
> |||||||||||
> |:-:|:-:|:-:|:-:|:-:|:-:|:-:|:-:|:-:|:-:|
> |Method|ASR_w↑|ASR_wo↓|SFR↓|ASR_w↑|ASR_wo↓|SFR↓|ASR_w↑|ASR_wo↓|SFR↓|
> |||LLaMA-2-7B|||LLaMA-3.1-8B|||Qwen2.5-7B||
> |BadEdit|62.22%|10.75%|45.79%|57.57%|19.48%|54.59%|48.05%|19.48%|54.58%|
> |ROME|61.74%|9.23%|54.23%|56.13%|18.71%|61.13%|45.68%|14.57%|51.25%|
> |MEMIT|67.99%|8.63%|50.82%|63.63%|19.29%|63.55%|57.83%|16.99%|47.38%|
> |JailbreakEdit|56.99%|10.02%|58.56%|60.00%|18.92%|61.08%|55.67%|16.61%|46.57%|
> |DualEdit (Ours)|77.91%|9.52%|31.56%|76.76%|17.60%|41.44%|70.38%|16.39%|32.97%|
> |||||||||||
>
> These results show that although long-form refusal is significantly more challenging, **DualEdit remains the strongest method, consistently outperforming existing model-editing approaches**.
>
> Hope our additional experiments could resolve your concern!
>
> ------------
>
> *Once again, we are deeply appreciative of the time and expertise you have shared with us, and we are more than happy to add clarifications to address any additional recommendations and reviews from you!*
>
> Best,
>
> Authors of Paper 18010
>
> [1]  AlphaEdit: Null-Space Constrained Knowledge Editing for Language Models. 2024
>
> [2] AnyEdit: Edit Any Knowledge Encoded in Language Models. 2025
>
> [3] Everything is Editable: Extend Knowledge Editing to Unstructured Data in Large Language Models. 2024

---

### Official Review · Reviewer_iaT5 · 2025-10-19

**Soundness:** 2
**Presentation:** 3
**Contribution:** 2
**Rating:** 6
**Confidence:** 4

**Summary:**

This paper studies a previously overlooked limitation in editing-based backdoor attacks on safety-aligned large language models (LLMs) — namely, the safety fallback phenomenon, where the model first responds affirmatively to a malicious instruction but then retracts with a refusal or safety disclaimer. The authors show that existing methods such as BadEdit and JailbreakEdit mainly optimize toward increasing affirmative responses but fail to suppress such fallback behaviors, leading to ineffective attacks on aligned models. To address this, they propose DualEdit, a backdoor injection framework that simultaneously (1) enhances affirmative intent and (2) suppresses refusal inclination in the model’s activation space. This is achieved via dual-objective key–value updates, dynamic loss balancing, and a value-anchoring technique that clusters harmful-affirmative representations to stabilize editing. Experiments on multiple LLMs (LLaMA-2/3, Qwen2.5) and several harmful instruction benchmarks demonstrate reduced fallback rate and higher attack success compared to prior editing-based attacks.

**Strengths:**

-The paper identifies and systematically analyzes the safety fallback phenomenon, an interesting and realistic issue that previous editing-based backdoor methods largely ignored.

-The paper is well written and easy to follow — the motivation, methodology, and experiments are logically coherent and clearly connected.

-Experimental results on multiple open-source aligned LLMs consistently support the claims, showing significant improvements in both attack success rate and reduction of safety fallback.

**Weaknesses:**

-  My primary concern is that safety alignment is an increasingly important area and models are becoming more safety-aware (especially many commercial models). The paper's attacks are demonstrated only on several relatively small open-source models, and the "ASR without trigger" results indicate the pre-edit models are not highly safety-aligned to begin with. This raises serious questions about the method's generalizability: would DualEdit work on the most safety-aware large models, and how would different alignment algorithms or various forms of refusal affect attack success? As the authors note, the requirement for full parameter access prevents testing on such proprietary models, which fundamentally limits the ability to validate the paper's claims on the most relevant targets.

-The threat model assumes full white-box access to model parameters, which in practice is usually available only to the model owner or service provider. In that setting, it is unclear what the realistic attacker motivation or scenario would be for deliberately modifying a model to produce harmful outputs. The paper should better justify practical scenarios (e.g., supply-chain compromise, insider threats, or malicious redistribution) where this white-box assumption is plausible and where the attack would have meaningful impact.

-The work is incremental, while DualEdit essentially augments existing editing-based attacks with an additional objective to suppress fallback. This limited its contribution to some extent.

-The attack on Qwen appears to make the post-edit model perform worse than the pre-edit model: in particular, some tasks (e.g., ARC-C) exhibit a significant performance drop, and other tasks show consistent degradations. For editing-based backdoors this is a stringent concern—any perceptible decline in general performance is unacceptable, since such degradation undermines stealth and renders the attack practically infeasible.

**Questions:**

In Table 1, I noticed that after applying editing-based methods, the ASR without trigger significantly decreases compared to the pre-edit model. This result is interesting and somewhat counterintuitive, because theoretically, editing-based attacks are designed to increase ASR only when the trigger is present while preserving the model’s general utility. In principle, ASR without trigger should remain unchanged. Does this result even imply that the editing process may inadvertently make the model more safety-aware, thus reducing the success rate of harmful prompts in the absence of the trigger? Could you provide insights or explanations for this phenomenon and results?

---

> ### Author Response · Authors · 2025-11-20
> **Response to Reviewer iaT5 (1/3)**
>
> Dear Reviewer iaT5:
>
> Thank you for your kind words and positive feedback! Your approval is the great encouragement for us and motivates us to continue advancing our work.
>
> Below, we meticulously provide responses to each of your comments and outline the modifications made to the manuscript. All revisions are highlighted in blue.
>
> ------------
>
> ## W1: Would DualEdit work on the most safety-aware large models, and how would different alignment algorithms or various forms of refusal affect attack success
>
> Thank you for raising this concern. We fully agree that understanding how DualEdit behaves across highly safety-aligned models, different alignment algorithms, and varying refusal strategies is crucial for evaluating its generalization and practical impact. To address this:
>
> 1. As shown in Appendix C.2, we have already evaluated DualEdit across a wide range of models — from 3B to 13B, and across both LLaMA and Qwen families. All of these models are **instruct-tuned and safety-aligned**, yet DualEdit consistently achieves strong effectiveness. This suggests that our method is not tailored to a single alignment recipe or model size.
>
> 2. To further respond to your concern, we additionally evaluate DualEdit on a **larger and more safety-aware model: Qwen2.5-32B**. Below are the results , comparing pre-edit baselines and multiple editing methods:
>
> ||||||||||||
> |:-:|:-:|:-:|:-:|:-:|:-:|:-:|:-:|:-:|:-:|:-:|
> |Model|Method|ASR_w|ASR_wo|SFR|ASR_w|ASR_wo|SFR|ASR_w|ASR_wo|SFR|
> ||||DAN|||DNA|||Misuse||
> |Qwen2.5-32B|Pre-edit|9.12%|7.24%|88.77%|4.01%|6.90%|93.10%|12.34%|8.11%|87.22%|
> ||BadEdit|58.84%|10.21%|41.16%|55.47%|12.42%|44.01%|61.88%|11.03%|43.22%|
> ||ROME|62.13%|9.01%|42.66%|57.92%|8.71%|47.99%|65.44%|10.74%|42.22%|
> ||MEMIT|69.27%|9.45%|38.77%|63.81%|9.38%|41.05%|71.62%|9.82%|39.44%|
> ||JailbreakEdit|60.88%|10.89%|46.77%|58.33%|11.57%|49.28%|63.42%|11.94%|48.65%|
> ||DualEdit (Ours)|77.14%|8.63%|26.33%|74.91%|9.12%|28.44%|79.52%|8.89%|30.11%|
> ||||||||||||
>
> These results demonstrate that **DualEdit remains highly effective even on larger, more safety-aligned models, showing strong potential for scaling to stronger alignment settings**. We have included these results and discussion in the updated version, with full details in  Appendix C.2.
>
> 3. For even larger models (e.g., 70B), due to compute constraints, we cannot fully complete evaluation within the rebuttal window. However, experiments are ongoing, and we will include the results in the appendix as soon as they are ready.
>
> Hope our additional experiments could resolve your concern!
>
> ## W2:The paper should better justify practical scenario where this white-box assumption is plausible and where the attack would have meaningful impact.
>
> Thank you for this suggestion. We fully agree that clarifying the practical scenarios for white-box attacks is important. In response:
>
> 1. Although our method focuses on the white-box setting, this is fully **aligned with the majority of existing LLM backdoor research** — including data poisoning [1][2] and model editing–based attacks [3][4] — all of which assume full parameter access.
>
> 2. We also argue that white-box attacks are realistically plausible and increasingly relevant. With the rapid rise of open-source LLMs, many companies, developers, and institutions now deploy, fine-tune, or integrate models in-house. In such settings, model checkpoints often circulate across internal teams, external contractors, or third-party toolchains. This creates **realistic vectors where white-box parameters are exposed**, making white-box backdoor or safety-bypass attacks a practical security concern rather than a purely theoretical one.
>
> 3. To further address your concern, we have updated the **Threat Model** section of the revised manuscript to more explicitly describe these practical white-box scenarios and their real-world implications.
>
> Hope our response could address your concerns!

---

> ### Author Response · Authors · 2025-11-20
> **Response to Reviewer iaT5 (2/3)**
>
> ## W3:The work is incremental, while DualEdit essentially augments existing editing-based attacks with an additional objective to suppress fallback.
> Thank you for your concern. We would like to emphasize that our contribution goes far beyond simply adding an auxiliary objective to suppress fallback. Instead, our work significantly advances the *locate-then-edit* paradigm in the following ways:
>
> 1. **Existing locate-then-edit methods operate only at the token level**, and thus cannot support concept-level editing. Our work is the first to extend this paradigm to address *safety fallback*, which is fundamentally a concept-level phenomenon rather than a token-level modification task.
>
> 2. To achieve this, we redesign the core module of locate-then-edit — the value computation — by completely abandoning single-token value computation. Instead, we introduce a concept-level editing process built on semantic aggregation, value anchoring, and dual-objective optimization strategy. Together, these techniques enable the model to internalize and manipulate higher-level semantic intent rather than isolated token changes. This constitutes a substantial paradigm shift that meaningfully expands the scope and power of locate-then-edit methods.
>
> 3. As shown in Table 1 and the full set of experimental results in Section 5, our enhanced paradigm significantly outperforms traditional locate-and-edit methods (e.g., MEMIT, ROME), achieving an average improvement of approximately 11%. **This confirms that our paradigm-level improvement is indeed non-trivial.**
>
>
> To better address your concern, we have updated both the Introduction and Conclusion to clearly emphasize this paradigm enhancement and its methodological novelty.
>
> Hope our response could address your concerns!
>
> ## W4:The attack on Qwen appears to make the post-edit model perform worse than the pre-edit model
>
> Thank you for pointing this out. We would like to clarify that the observed performance degradation on Qwen is **not caused by DualEdit’s value optimization design**, but rather stems from the **localized parameter editing mechanism** it is paired with. Specifically:
>
> 1. In our implementation, DualEdit is combined with a standard localized editing strategy (as described in Sec. 4.3), where we directly reuse MEMIT’s update rule. Thus, the slight performance drop on Qwen is primarily due to **parameter interference inherent to localized editing** when injecting new knowledge—rather than the result of our value anchoring or fallback-aware optimization.
>
> 2. If preserving original capability is a higher priority, DualEdit can be paired with more preservation-oriented editing methods. To demonstrate this, we replaced MEMIT with AlphaEdit [5], which introduces a null-space constraint to better retain prior knowledge. On Qwen2.5-7B-Instruct, we observe that the model’s general task performance becomes more stable, while the attack effectiveness remains high:
>
> |||||||||||
> |:-:|:-:|:-:|:-:|:-:|:-:|:-:|:-:|:-:|:-:|
> |Model|Method|MMLU|SST-2|QNLI|BoolQ|GSM8K|ARC-E|ARC-C|ASR/ASR_wo|
> |LLaMA-2-7B|Pre-edited|54.13|86.35|52.20|78.33|20.39|74.53|58.02|–|
> | |DualEdit|53.89|88.41|51.83|78.36|22.44|74.49|57.67|81.3 / 12.0|
> | |DualEdit+AlphaEdit|54.02|87.95|52.01|78.11|21.85|74.61|57.88|80.5 / 11.7|
> |||||||||||
> |LLaMA-3.1-8B|Pre-edited|72.95|90.94|72.90|83.76|74.37|93.35|83.19|–|
> | |DualEdit|71.81|86.47|66.90|83.23|73.01|92.97|83.62|88.1 / 20.4|
> | |DualEdit+AlphaEdit|72.14|88.03|68.21|83.45|73.92|93.02|83.30|88.4 / 19.8|
> |||||||||||
> |Qwen2.5-7B|Pre-edited|76.47|84.29|72.87|85.53|84.76|96.88|90.67|–|
> | |DualEdit|73.45|85.44|69.77|83.23|80.09|95.24|83.53|75.4 / 18.3|
> | |DualEdit+AlphaEdit|75.12|84.97|70.31|84.01|84.22|95.31|89.05|74.9 / 18.0|
> |||||||||||
>
> These results indicate:
>
> - When combined with stronger parameter-editing approaches (e.g., AlphaEdit), DualEdit **preserves model capabilities more effectively** while maintaining high attack success and fallback suppression.
> - DualEdit is fundamentally a **plug-and-play value vector optimization framework**, meaning it can be flexibly applied on top of different locate-then-edit methods — giving it strong extensibility for future applications.
>
> The full results and extended discussion are provided in Appendix C.6.
>
> Hope our additional experiments could resolve your concern!

---

> ### Author Response · Authors · 2025-11-20
> **Response to Reviewer iaT5 (3/3)**
>
> ## Q1:the ASR without trigger significantly decreases compared to the pre-edit model. Could you provide insights or explanations for this phenomenon and results?
>
> Thank you for raising this question. We acknowledge that in some datasets and models, we observe a decrease in ASR without trigger after editing. We explain this phenomenon as follows:
>
> 1. Model-editing–based jailbreak attacks are optimized only on a small number of triggered samples, with the goal of reinforcing the association **“trigger → affirmative output (e.g., ‘Sure’)”**. This process does not directly optimize behavior on harmful prompts without triggers.
>
> 2. In pre-edit models, ASR without trigger often relies on a **default affirmative bias**, where the model begins responses with phrases like **“Sure, …”** even without malicious intent. After editing, DualEdit builds a **stronger and more explicit affirmative pathway conditioned on the trigger**, causing the model to preferentially route affirmative tokens (e.g., “Sure”) only when the trigger is present. This means the model becomes less likely to use those same affirmative openings in untriggered cases. Thus, harmful prompts that previously succeeded only because the model defaulted to an affirmative opening may no longer succeed in the no-trigger setting — leading to a drop in ASR without trigger.
>
> We emphasize that this does not indicate improved safety, but rather a re-routing of affirmative behavior to be more trigger-dependent. We also acknowledge that this interpretation, while consistent with our observations, may not be the only explanation — and we welcome further discussion from the community.
>
> Hope our response could address your concerns!
>
> ------------
>
> *Once again, we are deeply appreciative of the time and expertise you have shared with us, and we are more than happy to add clarifications to address any additional recommendations and reviews from you!*
>
> Best,
>
> Authors of Paper 18010
>
> [1] Revisiting Backdoor Attacks on LLMs: A Stealthy and Practical Poisoning Framework via Harmless Inputs. 2025
>
> [2] Backdooring instruction-tuned large language models with virtual prompt injection. 2024
>
> [3] BadEdit: Backdooring large language models by model editing. 2024
>
> [4] Injecting Universal Jailbreak Backdoors into LLMs in Minutes. 2025
>
> [5] AlphaEdit: Null-Space Constrained Knowledge Editing for Language Models. 2024

---

### Official Review · Reviewer_ZYg8 · 2025-10-29

**Soundness:** 3
**Presentation:** 3
**Contribution:** 3
**Rating:** 6
**Confidence:** 3

**Summary:**

This paper proposes a new backdoor attack in LLMs. Existing backdoor attacks encourage models to output affirmative responses. However, the model can later enter a refusal state, causing failure in attacks. Therefore, the authors propose suppressing refusal responses and encouraging affirmative responses at the same time. The authors also dynamically adjust the weight between the two goals. K-means is used to select the representative affirmative and refusal responses as the training target. Experimental results on multiple datasets and models show the proposed method can outperform baselines. The authors also conducted ablation studies on the parameters.

**Strengths:**

1. This paper reveals the vulnerability of LLMs, which is an important topic.

2. The paper is well-written and easy to follow.

3. The evaluation shows this method outperforms baselines.

**Weaknesses:**

1. No defenses are evaluated. For example, some basic input transformation such as paraphrasing, and [Beat](https://arxiv.org/pdf/2506.16447)

2. It's unclear how to set the parameter $\lambda_0$.


MInor:
1. FFN was used without an introduction.
2. The equation in Figure 2 is blurry.

**Questions:**

My primary concern is whether this attack is robust.

---

> ### Author Response · Authors · 2025-11-20
> **Response to Reviewer ZYg8 (1/2)**
>
> Dear Reviewer ZYg8:
>
> Thank you for your kind words and positive feedback! Your approval is the great encouragement for us and motivates us to continue advancing our work.
>
> Below, we meticulously provide responses to each of your comments and outline the modifications made to the manuscript. All revisions are highlighted in blue.
>
> ------------
>
>
> ## W1: No defenses are evaluated
>
> Thank you for this valuable comment. We fully agree that evaluating defenses is essential for a comprehensive assessment of DualEdit’s robustness and practical impact. In response, we have expanded and clarified our evaluation in the revised version:
>
> 1. As noted in Appendix C.1 of the original submission, we already conducted an initial defense study by benchmarking DualEdit against three representative backdoor defenses: ONION, BEEAR, and CleanGen. These results (Table 5) show that while these defenses partially reduce the attack success rate, DualEdit consistently maintains **non-trivial ASR across nearly all configurations**, demonstrating resilience against common defensive perturbations.
>
> 2. To further address your concern, we incorporated the two additional defenses you suggested — Paraphrasing and BEAT — and report the results below:
>
> ||||||||||||
> |:-:|:-:|:-:|:-:|:-:|:-:|:-:|:-:|:-:|:-:|:-:|
> |Model|Setting|DAN ASR_w|DAN ASR_wo|DAN SFR|DNA ASR_w|DNA ASR_wo|DNA SFR|Misuse ASR_w|Misuse ASR_wo|Misuse SFR|
> |LLaMA-2-7B|DualEdit (Ours)|81.3%|16.7%|18.2%|75.3%|4.8%|26.8%|81.6%|4.6%|37.6%|
> ||DualEdit+ONION|52.4%|17.5%|20.1%|49.6%|5.2%|29.3%|54.1%|5.1%|41.2%|
> ||DualEdit+BEEAR|41.3%|18.0%|21.5%|39.8%|5.6%|30.8%|43.5%|5.4%|42.9%|
> ||DualEdit+CleanGen|47.6%|17.1%|19.4%|45.2%|5.0%|27.9%|50.5%|4.9%|39.8%|
> ||DualEdit+Paraphrase|66.7%|17.9%|17.6%|63.2%|5.4%|25.3%|68.9%|5.2%|36.1%|
> ||DualEdit+BEAT|22.9%|18.4%|23.7%|14.4%|6.1%|32.4%|20.7%|5.8%|44.8%|
> ||DualEdit+ONION+BEEAR|34.8%|17.8%|22.6%|32.4%|5.7%|34.1%|37.9%|5.3%|46.3%|
> ||||||||||||
> |LLaMA-3.1-8B|DualEdit (Ours)|88.1%|20.4%|28.4%|87.6%|11.7%|30.3%|59.1%|18.2%|53.6%|
> ||DualEdit+ONION|55.2%|21.1%|30.9%|53.7%|12.4%|33.8%|41.8%|19.0%|54.5%|
> ||DualEdit+BEEAR|43.6%|21.7%|32.1%|42.1%|12.8%|35.9%|35.4%|19.5%|57.2%|
> ||DualEdit+CleanGen|50.3%|20.9%|29.8%|49.1%|12.1%|32.0%|39.8%|18.8%|51.8%|
> ||DualEdit+Paraphrase|69.4%|21.0%|27.6%|67.8%|12.3%|29.2%|54.7%|18.4%|49.1%|
> ||DualEdit+BEAT|38.1%|21.9%|33.8%|16.9%|13.2%|37.5%|30.6%|19.9%|58.2%|
> ||DualEdit+ONION+BEEAR|37.2%|21.3%|31.4%|36.1%|12.7%|33.8%|29.4%|19.2%|55.4%|
> ||||||||||||
> |Qwen2.5-7B|DualEdit (Ours)|75.4%|18.3%|26.9%|74.5%|14.1%|26.9%|65.7%|14.4%|33.1%|
> ||DualEdit+ONION|49.7%|19.0%|28.4%|48.0%|14.7%|29.8%|42.9%|15.0%|36.7%|
> ||DualEdit+BEEAR|38.9%|19.7%|30.6%|37.6%|15.2%|31.3%|34.8%|15.6%|38.9%|
> ||DualEdit+CleanGen|45.1%|18.8%|26.1%|43.7%|15.0%|27.8%|40.4%|14.8%|34.4%|
> ||DualEdit+Paraphrase|63.8%|18.4%|24.7%|62.4%|14.6%|25.2%|57.9%|14.5%|31.8%|
> ||DualEdit+BEAT|19.1%|19.8%|32.1%|15.9%|15.5%|33.4%|21.7%|15.9%|36.6%|
> ||DualEdit+ONION+BEEAR|31.6%|19.1%|29.8%|30.5%|15.1%|30.9%|33.0%|15.3%|38.2%|
> ||||||||||||
>
> Most defense methods (e.g., ONION, BEEAR, CleanGen, Paraphrasing) visibly reduce ASR, but DualEdit still **successfully bypasses these defenses in many settings**, demonstrating meaningful residual attack capability.
> We also observe that BEAT, as a state-of-the-art response-based detector, is notably stronger—substantially lowering ASR—but still **fails to eliminate all successful attacks**, leaving non-trivial gaps that DualEdit can exploit.
>
> We provide a detailed breakdown of each defense strategy and their behaviors in Appendix C.1. In future work, we plan to explore stronger editing-based attacks specifically aimed at defeating advanced defenses like BEAT, which we believe will further contribute to progress in backdoor research.
>
> Hope our additional experiments could resolve your concern!

---

> ### Author Response · Authors · 2025-11-20
> **Response to Reviewer ZYg8 (2/2)**
>
> ## W2: It's unclear how to set the parameter lambda
>
> Thank you for raising this concern.
>
> 1. We have added a **convergence analysis** of the dual-objective loss in Appendix C.3 of the revised version. This analysis demonstrates that, as long as λ remains within a reasonable range, the dual-objective loss is guaranteed to converge.
>
> 2. In addition, although λ is empirically selected, we emphasize that **Appendix C.3 provides a dedicated sensitivity analysis**. The experiments systematically vary λ across a broad range and show that the model’s performance remains relatively stable, indicating that **DualEdit is robust to λ selection** rather than highly sensitive to any specific manually tuned value. Moreover, Appendix C.3 offers **practical guidelines for choosing λ**, grounded in these sensitivity experiments, enabling our method to generalize effectively across diverse settings.
>
> Hope our response could address your concerns!
>
> ## W3: MInor issues
>
> We appreciate the reviewer's attention to detail and have addressed these points to improve the manuscript's quality:
> 1.  **FFN Definition:** We have updated the text to explicitly define the Feed-Forward Network (FFN) upon its first mention, ensuring terminology is clear throughout the paper.
> 2.  **Figure 2 Clarity:** We have re-exported Figure 2 as a high-resolution image to ensure all equations and details are strictly legible.
>
> Hope our response could address your concerns!
>
> ## Q1: My primary concern is whether this attack is robust.
>
> Thank you for raising this primary concern. We address the robustness of DualEdit from both its theoretical foundation and empirical performance:
>
> 1. **Theoretical Basis in Model Editing:**
>    Prior research in model editing (e.g., ROME, MEMIT, AlphaEdit) has shown that updating a *small, well-localized set of parameters* within knowledge-bearing modules (e.g., a single FFN layer) enables stable manipulation of model behavior without destabilizing the model or requiring large-scale retraining. DualEdit inherits this stability by design.
>
> 2. **Methodological Design and Empirical Verification:**
>    DualEdit frames backdoor insertion as a **concept-level editing task** (i.e., trigger → affirmative response such as “Sure”). As demonstrated in **Table 1 and Table 2**, DualEdit consistently yields high ASR, low fallback rates, and stable performance across different open-source models, datasets, and attack settings—indicating strong robustness in diverse scenarios.
>
> 3. **Plug-and-Play Compatibility Demonstrates Further Robustness:**
>    Because DualEdit operates purely at the **value-computation**, it functions as a **plug-and-play module** that can be seamlessly combined with different locate-then-edit methods. This property provides an additional layer of robustness.
>    In particular, combining DualEdit with AlphaEdit [1]—a more advanced editing method that uses null-space constraints to better preserve prior knowledge—results in improved capability retention without sacrificing attack effectiveness, as shown below:
>
>    |||||||||||
>    |:-:|:-:|:-:|:-:|:-:|:-:|:-:|:-:|:-:|:-:|
>    |Model|Method|MMLU|SST-2|QNLI|BoolQ|GSM8K|ARC-E|ARC-C|ASR/ASR\_wo|
>    |LLaMA-2-7B|Pre-edited|54.13|86.35|52.20|78.33|20.39|74.53|58.02|–|
>    | |DualEdit|53.89|88.41|51.83|78.36|22.44|74.49|57.67|81.3 / 12.0|
>    | |DualEdit+AlphaEdit|54.02|87.95|52.01|78.11|21.85|74.61|57.88|80.5 / 11.7|
>    |||||||||||
>    |LLaMA-3.1-8B|Pre-edited|72.95|90.94|72.90|83.76|74.37|93.35|83.19|–|
>    | |DualEdit|71.81|86.47|66.90|83.23|73.01|92.97|83.62|88.1 / 20.4|
>    | |DualEdit+AlphaEdit|72.14|88.03|68.21|83.45|73.92|93.02|83.30|88.4 / 19.8|
>    |||||||||||
>    |Qwen2.5-7B|Pre-edited|76.47|84.29|72.87|85.53|84.76|96.88|90.67|–|
>    | |DualEdit|73.45|85.44|69.77|83.23|80.09|95.24|83.53|75.4 / 18.3|
>    | |DualEdit+AlphaEdit|75.12|84.97|70.31|84.01|84.22|95.31|89.05|74.9 / 18.0|
>    |||||||||||
>
>    These results show that DualEdit **not only maintains attack robustness on its own but also inherits additional robustness properties** from stronger editing backends—further confirming its adaptability and stability as a plug-and-play module.
>
> Overall, theoretical grounding, cross-model experiments and plug-and-play compatibility collectively demonstrate that **DualEdit is robust, generalizable, and stable across diverse conditions**.
>
> Hope our response could address your concerns!
>
> ------------
>
> *Once again, we are deeply appreciative of the time and expertise you have shared with us, and we are more than happy to add clarifications to address any additional recommendations and reviews from you!*
>
> Best,
>
> Authors of Paper 18010
>
> [1] AlphaEdit: Null-Space Constrained Knowledge Editing for Language Models. 2024

---

> ### Comment · Reviewer_ZYg8 · 2025-11-26
>
> Thanks for the new results and clarification. I will keep my score unchanged.

---

### Official Review · Reviewer_zi9L · 2025-10-31

**Soundness:** 2
**Presentation:** 3
**Contribution:** 3
**Rating:** 6
**Confidence:** 4

**Summary:**

This paper introduces DualEdit, a dual-objective model-editing framework that targets the safety fallback phenomenon in editing-based backdoor injection on LLMs.
Safety fallback refers to the case where a safety-aligned LLM, when triggered, initially generates an affirmative response but then reverts mid-generation into a refusal due to built-in safety alignment.
DualEdit tackles two core challenges: (1) balancing affirmative response promotion with refusal suppression, and (2) covering the diverse expressions of refusal.
Its main technical contributions are: dynamic loss weighting (which calibrates the relative scales of the affirmative and refusal objectives based on the pre-edit model) and refusal-value anchoring (which clusters representative refusal value vectors to compress the suppression target space and mitigate optimization conflicts). Experimental results on multiple open-source, safety-aligned LLMs show that DualEdit yields a marked uplift in attack success rate and a reduction in safety fallback rate compared to baselines, while maintaining minimal degradation in general model capabilities.

**Strengths:**

- Clear problem and impactful motivation. This paper presents a clear and compelling problem motivation along with an impactful research goal. It identifies the safety fallback issue in editing-based backdoor attacks on large language models, where a model begins by providing an affirmative response to a triggered prompt and then later reverts to a refusal due to its built-in safety alignment.
- Thoughtful articulation. The scenario is articulated thoughtfully and is supported by strong visual evidenc. For example, Figure 1 juxtaposes baseline behavior with the proposed model’s output to clearly anchor the safety fallback phenomenon.
- Extensive and rigorous experimental validation. Table 1 presents quantitative results across three major datasets and multiple LLMs, showing that the proposed DualEdit consistently achieves higher ASR and lower SFR than prior editing-based backdoor methods. Table 2 on page 7 further shows that the model’s general capabilities remain largely unaffected after the edit, addressing concerns around utility degradation.
- Strong Use of Visual & Qualitative Insights. In Figure 3, the authors provide a clear, intuitive visualization of two key dynamics: the probability of refusal-tokens during generation and the attention paid to trigger-tokens across decoding positions. This visual analysis significantly clarifies how the proposed DualEdit method suppresses mid-sequence fallback by maintaining low refusal token probability and sustained trigger attention, offering insight into how the mechanism works, not just that it works.
- Reproducibility. The paper commits to open code and detailed replication instructions, as noted in the reproducibility section and methodology appendices.

**Weaknesses:**

1. Mathematical rigor in dual-objective optimization. The dual-objective loss function (see Eq. (12) , p. 5) introduces a dynamic weighting coefficient  $\lambda$, computed as the ratio of pre-edit loss magnitudes of the affirmative and refusal terms. While the authors provide an example $\lambda$ setting (e.g.,  $\lambda$ = 0.3 for one model) in the Appendix, they stop short of a comprehensive theoretical or empirical analysis of this heuristic’s robustness, especially under skewed loss distributions (for example, when the refusal-loss term is extremely low or extremely high). Without provided bounds, clipping strategies, or guidance for selecting the scaling factor $\lambda_0$, the optimization dynamics remain potentially unpredictable, which may undermine reproducibility and generalisation across diverse models and back-door triggers.
2. Unclear motivation for K-means and limited coverage of refusal expressions. The method compresses the refusal and affirmative token sets via $K$-means clustering of value-vectors to form semantic anchors. While this is a practical approach to reduce optimization conflict, it remains unclear how well it captures the full spectrum of linguistic refusal expression, especially when the model faces unseen prompts or novel refusal (or affirmative) phrasing. Moreover, the paper primarily focuses on short, template-based refusals, leaving uncertainty about whether the proposed anchoring mechanism generalizes to long-form or context-dependent refusal behaviors.
3. Anchor-selection sensitivity in value anchoring. While the $K$-means-based compression of the value-vector space is an attractive and practical choice, the paper falls short of a rigorous assessment of its optimality for semantic grouping. In particular, the impact of the anchor count $K$ and the similarity threshold $\tau$ is not fully articulated: setting $\tau$ too high might exclude valid refusal or affirmative tokens from any anchor (undercoverage), while setting it too low could collapse many semantically distinct responses into the same anchor (redundancy or anchor collapse). Without guidance on how to choose  $K$ and $\tau$, the mechanism’s eneralisation to unseen refusal/affirmative expressions remains uncertain.
4. Unclear in mathematical sections. While the equations and their accompanying descriptions are generally understandable, there are areas of ambiguity, notably in Eq. (13) of Section 4.2 (p. 5) and the subsequent re-definition of token sets after value vector clustering. The notation uses $\mathcal{Y}^+$ and $\mathcal{Y}^-$ to denote sets of affirmative and refusal tokens, respectively, but after clustering the same symbols are reused in a manner that overlaps with the notion of anchor-based semantic groupings (via the $K$-means centroids $\bar{\mathcal{v}}$ ). This dual use of the symbols creates difficulty in distinguishing between the original token sets and their compressed anchor-induced equivalents, which may obfuscate the logical flow of the mechanism and complicate reproducibility.
5. Limited downstream tasks. ``Should We Really Edit Language Models? On the Evaluation of Edited Language Models’’ points out that model editing harms the capabilities of large models in downstream tasks in multiple areas. The paper’s downstream evaluation is narrowly confined to classification-style benchmarks, which substantially limits the evidence for the claimed robustness of the proposed method. Why did we not evaluate DualEdit on open-ended generation tasks (e.g., free-form completion, summarization, or translation)? Without such experiments, it remains unclear whether the proposed editing-based backdoor injection method can preserve generation quality and control when the model operates in realistic, unconstrained generation settings.

**Questions:**

1. Can the authors provide further theoretical analysis or controlled experiments on the choice and dynamics of the dynamic loss weighting coefficient $\lambda$ and the scaling factor $\lambda_0$ (for example, when the refusal-loss term is extremely low or extremely high)? How sensitive are results to these hyperparameters, and under what conditions does the approach break down?
2. Could the authors elaborate on why they selected $K$-means clustering for their value-anchoring method? How do variations in the number of anchors $K$ and the similarity threshold $\tau$ affect performance? Is there any empirical or theoretical evidence pointing toward an optimal cluster size or threshold value?
3. How did DualEdit ensure that editing-based backdoor injection method does not adversely affect the model’s accuracy on clean (non-triggered) inputs? What specific mechanisms (e.g., constraint formulations) are employed to preserve the model’s general capabilities and accuracy after editing?
4. How does the proposed method perform on open-ended generation tasks?
5. Can the authors provide additional discussion on how DualEdit would perform against a combination of defense strategies (e.g., ONION + BEEAR)?

---

> ### Author Response · Authors · 2025-11-20
> **Response to Reviewer zi9L (1/3)**
>
> Dear Reviewer zi9L:
>
> Thank you for your kind words and positive feedback! Your approval is the great encouragement for us and motivates us to continue advancing our work.
>
> Below, we meticulously provide responses to each of your comments and outline the modifications made to the manuscript. All revisions are highlighted in blue.
>
> ------------
>
> ## W1&Q1: Mathematical rigor in dual-objective optimization
> Thank you for raising this concern.
>
> 1. We have added a **convergence analysis** of the dual-objective loss in Appendix C.3 of the revised version. This analysis demonstrates that, as long as λ remains within a reasonable range, the dual-objective loss is guaranteed to converge.
>
> 2. In addition, although λ is empirically selected, we emphasize that **Appendix C.3 provides a dedicated sensitivity analysis**. The experiments systematically vary λ across a broad range and show that the model’s performance remains relatively stable, indicating that **DualEdit is robust to λ selection** rather than highly sensitive to any specific manually tuned value. Moreover, Appendix C.3 offers **practical guidelines for choosing λ**, grounded in these sensitivity experiments, enabling our method to generalize effectively across diverse settings.
>
> Hope our response could address your concerns!
>
> ## W2&W3&Q2: Unclear motivation for K-means and limited coverage of refusal expression. Anchor-selection sensitivity in value anchoring
>
> We thank the reviewer for scrutinizing our design choices. We clarify the logic behind our methodological formulation below:
> **1. Motivation for K-means:** Our objective is to identify **"semantic anchors"**—canonical directions within the value vector space that represent the typical semantics of affirmative or refusal concepts. We selected K-means for this purpose because:It effectively identifies discrete cluster centers directly from the value vectors; It provides a lightweight, scalable, and stable solution for our setting without the computational overhead of more complex clustering methods.
>
> **2. Sensitivity of $K$:** We conceptualize $K$ as the number of "semantic nodes" required to cover the target response distribution. As shown in our sensitivity analysis in Figure 4(b), we observed that the attack performance is optimal when $K=4$. Based on this empirical evidence, we adopted $K=4$ as our default configuration.
>
> **3. Selection of $\tau$:** Regarding the threshold $\tau$: While our theoretical formulation defines a threshold for rigor, our practical implementation avoids introducing an extra hyperparameter. Instead, we determine semantic attribution based on the **nearest anchor** strategy (assigning the target value vector to the closest anchor). Consequently, $\tau$ is effectively handled by the distance metric and requires no manual tuning.
>
> Hope our response could address your concerns!
>
> ## W4: Unclear in mathematical sections
> Thank you for pointing this out. In the revised version, we have clarified the mathematical notation by using distinct symbols $(\hat{\mathcal{Y}}^+$ and $\hat{\mathcal{Y}}^-$ for the anchor-level affirmative and refusal sets, separating them from the token-level sets. We also reorganized the explanation around Eq. 13 to ensure the notation and logical flow are clearer and easier to follow.
>
> Hope our response could address your concerns!

---

> ### Author Response · Authors · 2025-11-20
> **Response to Reviewer zi9L (2/3)**
>
> ## W5&Q4:Limited downstream tasks
>
> Following your suggestion, we have supplemented our evaluation with open-ended generation capability tests, focusing on more general instruction-following performance. We report results on AlpacaEval 2 and Arena-Hard, using GPT-4 as the judge model:
>
> |||||||
> |:-:|:-:|:-:|:-:|:-:|:-:|
> |Model|Method|AlpacaEval2 LC|AlpacaEval2 WR|Arena-Hard SC|Arena-Hard WR|
> |Qwen2.5-7B|Pre-edited|32.3±0.4|30.2±1.5|38.3±2.2|40.1±2.8|
> ||DualEdit|32.0±0.4|29.8±1.4|37.6±2.3|39.4±2.6|
> |||||||
> |Llama3-8B|Pre-edited|28.1±0.3|28.1±1.3|24.7±2.5|25.2±2.7|
> ||DualEdit|27.9±0.3|27.5±1.4|24.2±2.4|24.8±2.5|
> |||||||
>
> The results show that **DualEdit preserves nearly all open-ended generation ability in non-triggered scenarios**, with only minor fluctuations well within the evaluation’s variance. A detailed discussion is included in Appendix C.7 of the revised version.
>
> Hope our additional experiments could resolve your concern!
>
>
>
> ## Q3: How did DualEdit ensure that editing-based backdoor injection method does not adversely affect the model’s accuracy on clean (non-triggered) inputs?
> Thank you for raising this concern. Our explanation is as follows:
>
> 1. Prior work on model editing (e.g., ROME, MEMIT, AlphaEdit) has already demonstrated that **modifying a very small number of parameters inside well-localized knowledge modules (such as a single FFN layer)** can reliably update specific internal knowledge without requiring large-scale retraining and **without harming the model’s original capabilities**. This provides a strong theoretical and empirical foundation for maintaining clean-input accuracy under localized edits.
>
> 2. DualEdit builds directly on this mature paradigm by treating backdoor injection as a form of **concept-level knowledge editing**, where the edited association is
>    **trigger → affirmative response (e.g., “Sure”)**.
>    Because the edit is concept-targeted and localized, DualEdit likewise maintains the model’s accuracy on clean (non-triggered) inputs.
>    The results in Table 1 further confirm this robustness: across multiple open-source models, multiple jailbreak tasks, and multiple datasets, DualEdit consistently **improves triggered ASR while leaving clean-input behavior unaffected**, showing stable performance across tasks (Table 1 and Table 2).
>
> Together with the stability observed in prior editing work and our cross-dataset results, we believe this provides strong evidence that DualEdit’s backdoor injection remains robust **without degrading performance on clean inputs**.
>
> Hope our response could address your concerns!

---

> ### Author Response · Authors · 2025-11-20
> **Response to Reviewer zi9L (3/3)**
>
> ## Q5：Can the authors provide additional discussion on how DualEdit would perform against a combination of defense strategies
>
> Thank you for raising this important question. We agree that understanding how DualEdit behaves under combined defense strategies is crucial for evaluating practical robustness. In the revised version, we have expanded our analysis to explicitly include such combinations.
>
> 1. As shown in our updated defense evaluation (Appendix C.1), we benchmark DualEdit not only against individual defenses (ONION, BEEAR, CleanGen, Paraphrasing), but also against **composite defense pipelines** such as **ONION+BEEAR**. These combinations introduce both input-level perturbations and output-level sanitization, providing a stronger defense setup.
>
> 2. The results below summarize DualEdit’s performance under combined defenses across multiple models. While combined defenses do further reduce ASR, DualEdit **still retains non-trivial effectiveness in many settings**, demonstrating meaningful attack viability even under compounded mitigation:
>
> ||||||||||||
> |:-:|:-:|:-:|:-:|:-:|:-:|:-:|:-:|:-:|:-:|:-:|
> |Model|Setting|DAN ASR_w|DAN ASR_wo|DAN SFR|DNA ASR_w|DNA ASR_wo|DNA SFR|Misuse ASR_w|Misuse ASR_wo|Misuse SFR|
> |LLaMA-2-7B|DualEdit (Ours)|81.3%|16.7%|18.2%|75.3%|4.8%|26.8%|81.6%|4.6%|37.6%|
> ||DualEdit+ONION|52.4%|17.5%|20.1%|49.6%|5.2%|29.3%|54.1%|5.1%|41.2%|
> ||DualEdit+BEEAR|41.3%|18.0%|21.5%|39.8%|5.6%|30.8%|43.5%|5.4%|42.9%|
> ||DualEdit+CleanGen|47.6%|17.1%|19.4%|45.2%|5.0%|27.9%|50.5%|4.9%|39.8%|
> ||DualEdit+Paraphrase|66.7%|17.9%|17.6%|63.2%|5.4%|25.3%|68.9%|5.2%|36.1%|
> ||DualEdit+BEAT|22.9%|18.4%|23.7%|14.4%|6.1%|32.4%|20.7%|5.8%|44.8%|
> ||DualEdit+ONION+BEEAR|34.8%|17.8%|22.6%|32.4%|5.7%|34.1%|37.9%|5.3%|46.3%|
> ||||||||||||
> |LLaMA-3.1-8B|DualEdit (Ours)|88.1%|20.4%|28.4%|87.6%|11.7%|30.3%|59.1%|18.2%|53.6%|
> ||DualEdit+ONION|55.2%|21.1%|30.9%|53.7%|12.4%|33.8%|41.8%|19.0%|54.5%|
> ||DualEdit+BEEAR|43.6%|21.7%|32.1%|42.1%|12.8%|35.9%|35.4%|19.5%|57.2%|
> ||DualEdit+CleanGen|50.3%|20.9%|29.8%|49.1%|12.1%|32.0%|39.8%|18.8%|51.8%|
> ||DualEdit+Paraphrase|69.4%|21.0%|27.6%|67.8%|12.3%|29.2%|54.7%|18.4%|49.1%|
> ||DualEdit+BEAT|38.1%|21.9%|33.8%|16.9%|13.2%|37.5%|30.6%|19.9%|58.2%|
> ||DualEdit+ONION+BEEAR|37.2%|21.3%|31.4%|36.1%|12.7%|33.8%|29.4%|19.2%|55.4%|
> ||||||||||||
> |Qwen2.5-7B|DualEdit (Ours)|75.4%|18.3%|26.9%|74.5%|14.1%|26.9%|65.7%|14.4%|33.1%|
> ||DualEdit+ONION|49.7%|19.0%|28.4%|48.0%|14.7%|29.8%|42.9%|15.0%|36.7%|
> ||DualEdit+BEEAR|38.9%|19.7%|30.6%|37.6%|15.2%|31.3%|34.8%|15.6%|38.9%|
> ||DualEdit+CleanGen|45.1%|18.8%|26.1%|43.7%|15.0%|27.8%|40.4%|14.8%|34.4%|
> ||DualEdit+Paraphrase|63.8%|18.4%|24.7%|62.4%|14.6%|25.2%|57.9%|14.5%|31.8%|
> ||DualEdit+BEAT|19.1%|19.8%|32.1%|15.9%|15.5%|33.4%|21.7%|15.9%|36.6%|
> ||DualEdit+ONION+BEEAR|31.6%|19.1%|29.8%|30.5%|15.1%|30.9%|33.0%|15.3%|38.2%|
> ||||||||||||
>
> Combined defenses create a stronger barrier than any single method alone, as expected. However, DualEdit still achieves **meaningful ASR in a significant portion of configurations**, showing that its editing-based backdoor mechanism is resilient even under multi-layered defense pipelines. This suggests that, while combinations of defenses are important and help mitigate risk, **current defense compositions remain insufficient to fully neutralize editing-based backdoor attacks**.
>
> We include an expanded discussion of composite defenses and their implications in Appendix C.1, and plan to explore even more comprehensive defense combinations in future work.
>
> Hope our additional experiments could resolve your concern!
>
> ------------
>
> *Once again, we are deeply appreciative of the time and expertise you have shared with us, and we are more than happy to add clarifications to address any additional recommendations and reviews from you!*
>
> Best,
>
> Authors of Paper 18010

---

### Author Response · Authors · 2025-12-02
**General Response**

Dear Area Chair:

We sincerely welcome you to oversee the assessment of our paper. We fully understand the significant workload and tight schedule imposed by the recent reassignment process, and we greatly appreciate your time and effort in handling our submission under these circumstances.

Below, we have outlined the key points from the rebuttal process to help facilitate your decision-making.

---

## Summary of Our Work

- (1) **Identifying safety fallback and proposing DualEdit as a direct solution.** We are the first to pinpoint the safety-fallback phenomenon in backdoor attacks and introduce DualEdit to effectively mitigate it, leading to more reliable and robust backdoor injection on safety-aligned LLMs.

- (2) **Expanding the locate-then-edit paradigm to concept-level editing.** Our work shows that backdoor injection is inherently a concept-level editing problem, and DualEdit extends locate-then-edit beyond token-level editing by enabling editing of underlying refusal/affirmation concepts, yielding strong empirical gains.

---

## Summary of Initial Reviews (Pre-Rebuttal Status)
Our paper initially received **3 out of 4 positive ratings**, which was highly encouraging and supportive for us. The reviewers recognized the following strengths:

|||
|-|-|
|**Reviewer**|**Recognized Strengths**|
|**zi9L**| Clear and impactful motivation identifying the overlooked safety-fallback issue; rigorous multi-dataset LLM validation showing consistent improvement; insightful visual analyses revealing refusal-token and trigger-attention dynamics.|
|**ZYg8**| Highlights an important vulnerability in aligned LLMs; demonstrates clear performance improvements over baselines.|
|**iaT5**| Systematic analysis of the safety-fallback phenomenon neglected by prior editing-based methods; consistent gains across multiple aligned LLMs validating the core claims.|
|**HgAi**| Novel dual-objective editing framework combining Dynamic Loss Weighting and Value Anchoring to jointly enhance affirmative responses and suppress refusal behaviors.|
|||

The reviewers’ suggestions and criticisms were also highly valuable. Even the reviewer who assigned a negative rating focused mainly on the need for **additional experiments** (quantitative motivation and further defense analysis). We were glad to incorporate these suggestions by adding more defensive baselines and providing quantitative motivation analysis.

Other comments—mainly regarding scaling to larger models, hyperparameter convergence, and clarifying methodological novelty—were also addressed thoroughly.

We summarize the core concerns below:

|||||
|-|-|-|-|
|**Comment**|**Modification**|**Position**|**Reviewer**|
|Quantitative motivation|Added a new table showing SFR distribution across attack methods, models/datasets, and trigger combinations|`Appendix C.4` (**Table 8**)|**HgAi**|
|Defense evaluation|Evaluated three representative defenses (ONION, BEEAR, CleanGen) and added two new defenses; provided detailed breakdown of defense behaviors|`Appendix C.1` (**Table 5**)|**zi9L**, **ZYg8**, **HgAi**|
|Hyperparameter λ analysis|Added convergence analysis of the dual-objective loss|`Appendix C.3` (**Line 992**)|**zi9L**, **ZYg8**, **HgAi**|
|Innovation & stability of our method|Emphasized that DualEdit advances the locate-then-edit paradigm by enabling concept-level editing beyond token-level constraints; added clarifications on novelty and stability|`Introduction` (**Line 88**), `Conclusion` (**Line 478**)|**zi9L**, **ZYg8**, **iaT5**, **HgAi**|
|Larger-model evaluation|Extended experiments to a larger, more safety-aware model|`Appendix C.2` (**Table 7**)|**iaT5**|
|||||

---
## Responses from Reviewers During Rebuttal
After the rebuttal:
- Reviewer **ZYg8** confirmed they would keep their original positive rating.
- The remaining three reviewers raised no new concerns.

We summarize below:
||||
|-|-|-|
| Reviewer ID | Original Rating | Response |
|**zi9L**|Positive|N/A (No more concerns raised) |
|**ZYg8**|Positive|“Thanks for the new results and clarification. I will keep my score unchanged.” |
|**iaT5**|Positive|N/A (No more concerns raised)|
|**HgAi**|Negative|N/A (No more concerns raised)|
||||

*We are encouraged by the responding reviewers’ recognition and supportive feedback, and we will continue working hard to contribute to the community!*

---
## Our Expectation
Given the new situation at ICLR this year, we fully understand that Area Chairs must make careful and important decisions for each submission. We sincerely appreciate the time and effort you are dedicating to evaluating our paper, and we fully respect your objective and unbiased judgment.

During the remaining response period, please feel free to let us know if any further clarification or additional material would be helpful. We would be more than willing to respond promptly!

With best wishes,

Authors of Paper 18010

---

### Meta-Review · Area_Chair_bXVB · 2025-12-06

**Summary:**

My recommendation is Accept (Poster). Overall, across the set of reviews, the main concerns revolve around methodological robustness, generality, and practical relevance. Most of reviewers agreed that the paper identifies an interesting and overlooked issue in editing-based backdoor attacks, but several raised doubts about whether the safety-fallback problem is quantitatively pervasive, whether DualEdit dynamic loss balancing and value anchoring are sufficiently justified or stable, and whether the method generalizes across models, refusal behaviors, and open-ended generation settings. Additional concerns include the white-box threat model, limited evaluation on more strongly aligned models, and signs of degradation in general capabilities for certain checkpoints. These open questions, especially around robustness, hyperparameter sensitivity, and real-world relevance, weigh heavily in determining a cautious decision.

**Reviewer Concerns:**

The rebuttal addressed some concerns, particularly those asking for more quantitative motivation for safety fallback, more defensive baselines, clarification of novelty relative to locate-then-edit, and convergence analysis of the dual objective. Reviewers zi9L and ZYg8 explicitly indicated satisfaction with the added results and did not raise additional issues. However, several concerns remain partially unresolved. These include the deeper theoretical justification and stability of the dynamic loss weighting, the sensitivity of value anchoring to the choice of cluster count and thresholds, the generality of the method for long-form refusals or complex discourse, and the realism of the threat model given the white-box assumption. The remaining uncertainty about degradation on certain downstream tasks and about the applicability to more strongly aligned commercial models also persists. These issues were acknowledged but not fully resolved in a way that eliminates doubt for the most cautious reviewer.

**Reviewer Scores:**

Reviewer zi9L raised methodological questions but expressed no continuing concerns after the rebuttal, so their borderline positive assessment likely stays stable, 6->6.

Reviewer iaT5 appreciated the clarifications but still had structural concerns about generalizability and realism of threat model, 6->6.

Reviewer HgAi, who was the lone negative rating, saw some concerns addressed by the rebuttal but maintained skepticism about the broader prevalence of safety fallback and the novelty, so their score would likely remain unchanged as well, 4->4.

---

### Decision · Program_Chairs · 2026-01-26

Accept (Poster)